# Parecoxib Enhances Resveratrol against Human Colorectal Cancer Cells through Akt and TXNDC5 Inhibition and MAPK Regulation

**DOI:** 10.3390/nu16173020

**Published:** 2024-09-06

**Authors:** Wan-Ling Chang, Kai-Chien Yang, Jyun-Yu Peng, Chain-Lang Hong, Pei-Ching Li, Soi Moi Chye, Fung-Jou Lu, Ching-Wei Shih, Ching-Hsein Chen

**Affiliations:** 1Department of Anesthesiology, Chang Gung Memorial Hospital at Chiayi, No. 8, West Section of Jiapu Road, Chiayi County, Puzi City 613016, Taiwan; chjack1975@yahoo.com.tw (W.-L.C.); pengjyun550@gmail.com (J.-Y.P.); leisure@cgmh.org.tw (C.-L.H.); peiching@cgmh.org.tw (P.-C.L.); 2Department and Graduate Institute of Pharmacology, College of Medicine, National Taiwan University, No. 1, Jen Ai Road Section 1, Taipei 100233, Taiwan; kcyang@ntu.edu.tw; 3School of Health Science, Division of Applied Biomedical Science and Biotechnology, IMU University, Bukit Jalil, Kuala Lumpur 57000, Malaysia; chye_soimoi@imu.edu.my; 4Institute of Medicine, Chung Shan Medical University, No. 110, Section 1, Jianguo North Road, Taichung City 402306, Taiwan; fjlu@csmu.edu.tw; 5Department of Microbiology, Immunology and Biopharmaceuticals, College of Life Sciences, National Chiayi University, A25-303 Room, Life Sciences Hall, No. 300, Syuefu Road, Chiayi City 600355, Taiwan; a0958798219@gmail.com

**Keywords:** parecoxib, resveratrol, TXNDC5, Akt, MAPK, apoptosis, colorectal cancer

## Abstract

In this study, we discovered the mechanisms underlying parecoxib and resveratrol combination’s anti-cancer characteristics against human colorectal cancer DLD-1 cells. We studied its anti-proliferation and apoptosis-provoking effect by utilizing cell viability 3-(4,5-Dimethylthiazol-2-yl)-2,5-diphenyltetrazolium bromide (MTT) assay, fluorescence microscope, gene overexpression, Western blot, and flow cytometry analyses. Parecoxib enhanced the ability of resveratrol to inhibit cell viability and increase apoptosis. Parecoxib in combination with resveratrol strongly enhanced apoptosis by inhibiting the expression of thioredoxin domain containing 5 (TXNDC5) and Akt phosphorylation. Parecoxib enhanced resveratrol-provoked c-Jun *N*-terminal kinase (JNK) and p38 phosphorylation. Overexpression of TXNDC5 and repression of JNK and p38 pathways significantly reversed the inhibition of cell viability and stimulation of apoptosis by the parecoxib/resveratrol combination. This study presents evidence that parecoxib enhances the anti-cancer power of resveratrol in DLD-1 colorectal cancer cells via the inhibition of TXNDC5 and Akt signaling and enhancement of JNK/p38 MAPK pathways. Parecoxib may be provided as an efficient drug to sensitize colorectal cancer by resveratrol.

## 1. Introduction

“Dietary phytochemicals” refer to compounds in fruits, vegetables, grains, and other foods. They are different from vitamins and minerals. Phytochemicals are “non-nutrients”—compounds which are not essential to the organism, but may play a very beneficial role to the body, including prevention of non-communicable diseases, like cardiac vascular diseases, cancer, diabetes, etc. These chemicals have become hot topics for medical research. What excites scientists most is that the dietary phytochemicals in plant foods have a remarkably ability to inhibit many cancer cells [1,2,3]. As a result, scientists are working to uncover the functions of dietary phytochemicals in plant foods to provide evidence for cancer prevention.

Resveratrol (3,4′,5-trihydroxy-*trans*-stilbene) is generated by certain plants including numerous dietary sources such as peanuts, grapes, blueberries, raspberries, apples, plums, and products derived from them (e.g., wine) [4]. Resveratrol can be isolated and purified from these biological sources or synthesized in a few steps with an overall high yield [4]. The beneficial intake of resveratrol is between 30 and 150 mg [5]. A dietary evaluation was performed to show that the daily resveratrol intake by Chinese people through everyday foods was only 0.783 mg, which was significantly lower than the beneficial dose [5]. Among the chief food kinds, fruits appeared as the primary source of resveratrol, contributing to 88.35% of the total intake [5]. Resveratrol-enriched supplements might be appropriate to permit a daily intake of therapeutically related doses (currently presumed to be 1 g) that are not offered by beverages or conventional foods [4]. Extensive metabolism in the liver and intestine causes an oral bioavailability considerably lower than 1% [6]. Metabolic studies, both in urine and in plasma, have uncovered the major metabolites of resveratrol to be glucuronides and sulfates [6]. Other study demonstrates that no phase I metabolites were detected, but the phase II conjugates resveratrol-3-glucuronide and resveratrol-3-sulfate was found based on LC-MS and LC-MS-MS analysis and comparison with synthetic standards. Although these data point to resveratrol diffusing quickly across the intestinal epithelium, broad phase II metabolism during absorption might decrease resveratrol bioavailability [6,7]. The major sites of resveratrol metabolism include the intestine and liver [6]. There is no first passing effect. The resveratrol is absorbed in the small intestine and sent to the liver for phase II enzyme metabolism and to produce glucuronides and sulfates of resveratrol [6,7]. Resveratrol occurs in the form of two geometric isomers—*trans* and *cis*—and only the *trans* geometric isomer shows biological activity [8]. As a natural phytoalexin, resveratrol is a secondary metabolite derived from plant resistance to pathogenic attack and environmental stress [9]. It was first separated from the roots of *Veratrum album* in 1940 and was later isolated from the roots of *Polygonum cuspidatum* in 1963 [10]. Resveratrol is a dietary phytochemical with potential to improve cancer therapy and it indicates advances in cancer therapy. Resveratrol can induce DNA damage-mediated senescence in breast and liver cancer cells [11]. It has gained increasing interest because of its anti-cancer activities, low toxicity, proliferation suppression [12], apoptosis stimulation [13], autophagic death [14], and anti-metastasis effect [15]. The anti-cancer effects of resveratrol have been demonstrated in various cancer types. In colorectal cancer, resveratrol could inhibit the TGF-β-stimulated epithelial to mesenchymal transition and decrease the metastatic rate of lung and liver [16]. In breast cancer, resveratrol has the potential to suppress cell growth, and can induce apoptosis by increasing cytochrome c release, Bax/Bcl-xL ratio, and the cleavage of caspase 3 and PARP [13]. In human pancreatic cancer cells, resveratrol could induce apoptotic cell death by downregulating the anti-apoptotic protein MCL-1 [17]. Research shows that resveratrol has multiple anti-cancer mechanisms. It can promote apoptosis, which causes cancer cells to die on their own, and prevents their growth. In addition, resveratrol can protect white blood cells and endothelial cells from death due to oxidative stress caused by chemotherapeutical drugs or radiation therapy. Hence, resveratrol has potential value in inhibiting cancer cell growth.

The first-generation non-steroid anti-inflammatory drugs (NSAIDs) are mainly non-selective to inhibit both COX-1 and COX-2. Constitutive COX-1 is considered to mediate prostaglandin-dependent gastric protection [18]. COX-1 can assist the secretion of the gastric wall mucosa and prevent gastric acid from eroding the gastric wall mucosa [18]. Therefore, first-generation NSAIDs often have the serious side effect of gastric ulcers [19]. Selective inhibition of COX-2 is so important in comparison to the first generation NSAIDs, because gastric ulcers rarely occur. One of the serious side effects provided by COX-2 inhibitors is an increased risk of suffering myocardial infarction and death [20]. Rofecoxib was withdrawn from the market for this reason, but the similar COX-2 selective etoricoxib has replenished it in Europe but not in the United States [20]. Parecoxib is a generally prescribed analgesic and antipyretic drug and has an excellent selective inhibitory function on cyclooxygenase (COX)-2. Parecoxib can manage opioid-induced hyperalgesia [21]. Parecoxib diminishes postsurgical pain and accelerates movement more than the controlled analgesia of a patient [22]. Parecoxib can multimodal preemptive analgesia in reducing postoperative acute pain in hip and knee replacement patients, and decrease cumulative opioid consumption without increasing the risk of adverse drug events [23]. Parecoxib protects against myocardial ischemia/reperfusion via targeting the PKA-CREB signaling pathway [24]. Our previous investigations demonstrated that parecoxib possesses an anti-metastasis function by inhibiting the epithelial to mesenchymal transition and the Wnt/β-catenin signaling route in DLD-1 human colorectal cancer cells [25]. Parecoxib decreases glioblastoma migration, invasion, and cell proliferation by upregulating miRNA-29c [26]. Moreover, parecoxib combined with sufentanil inhibits the metastasis and proliferation of HER2-positive breast cancer cells by regulating the epithelial to mesenchymal transition [27], representing a potential synergistic antitumor effect. Recently, we demonstrated that parecoxib and 5-fluorouracil synergistically reduce the epithelial to mesenchymal transition and subsequent metastasis in colorectal cancer through targeting PI3K/Akt/NF-κB signaling [28]. These studies illustrate the potential of parecoxib to assist anti-cancer agents in enhancing anti-cancer effects.

TXNDC5 is a disulfide isomerase predominantly expressed in the endoplasmic reticulum [29]. Evidence indicates that TXNDC5 is upregulated by hypoxia in tumor endothelium and endothelial cells [30]. TXNDC5 is upregulated in several cancer types, such as lung, liver, esophageal, stomach, breast, uterine, and cervical carcinoma [31]. Previous studies validated that colorectal cancer tissues overexpress TXNDC5, and this overexpression is associated with poor clinical pathological characters in vivo. Moreover, an in vitro study showed that hypoxia provokes TXNDC5 production by upregulating HIF-1α; this consequence may elevate the survival and proliferation of colorectal cancer cells in a hypoxia environment. Under hypoxia, TXNDC5 performs as a prominent stress survival factor to induce the tumorigenesis of colorectal cancer by adjusting hypoxia-induced ROS/ER stress signaling [32]. However, no studies have evaluated the role of TXNDC5 or its related mechanisms on the combination of parecoxib and resveratrol on human colorectal cancer cells.

Akt, a proto-oncogene, belongs to the serine/threonine kinase family that adjusts downstream mediators and handles crucial metabolic procedures and cell survival [33]. Moreover, Akt stimulates cell cycle progress and prevents apoptosis. About 60–70% of Akt is highly activated in human colon cancers [34]. Targeting Akt signaling, from the viewpoint of discovering innovative molecular targets for cancer therapy, has revealed therapeutic inhibitors and candidates in critical pathways. The important characteristics for Akt kinase have provided an excellent target for acquiring therapeutic drugs for cancer [35].

Mitogen-activated protein kinase (MAPK) pathways are recognized kinase components that are related to cancer and have a critical function in signal transduction from the environment to the cell for controlling essential cellular progression processes, including differentiation, proliferation, apoptosis, and migration [36]. MAPKs contain three subgroups: JNK, p38, and extracellular signal-regulated kinase (ERK) [37]. Therefore, the discovery of new drugs to target MAPK pathways that can increase the capacity of chemotherapy to act on colorectal cancer homeostasis is noteworthy, because it may offer beneficial clinical consequences, enhance life quality during therapy, and diminish the side effects of chemotherapy.

This study investigated the synergistic anti-cancer effects of parecoxib and resveratrol combination on the cellular viability and apoptosis of human colorectal cancer cells at the cellular level and explored the underlying mechanisms. These findings may uncover the anti-cancer potential of resveratrol joined with clinically practicable concentrations of parecoxib in the therapy of colorectal cancer. This work offers a theoretical foundation for the clinical choice of a suitable chemotherapeutic regimen.

## 2. Materials and Methods

### 2.1. Reagents and Chemicals

RPMI-1640 medium was purchased from Hyclone (South Logan, UT, USA). Fetal bovine serum (FBS) and penicillin streptomycin–glutamine was obtained from Gibco Inc. (Freehold, NJ, USA). Resveratrol (3,4′,5-Trihydroxy-*trans*-stilbene), MTT, dimethyl sulfoxide (DMSO), crystal violet, trypan blue, 4,6-diamidino-2-phenylindole (DAPI), dichlorodihydrofluorescein diacetate (DCFH-DA), and other chemicals were acquired from Sigma-Aldrich Corp. (St. Louis, MO, USA). Parecoxib was supplied by Pfizer (Sydney, NSW, Australia). The Bio-Rad protein assay kit was bought from Bio-Rad Laboratories (Richmond, CA, USA). X-tremeGENE™ HP DNA transfection reagent was provided by Roche (Raleigh, NC, USA). Primary antibodies against p53, Bax, PARP, p-Akt, GAPDH and tubulin were bought from Santa Cruz Biotechnology, Inc. (Santa Cruz, CA, USA). Primary antibodies against TXNDC5 were purchased from Abcam (Waltham, MA, USA). The pLAS2w.pPuro and pLAS2w.pPuro-hTXNDC5 vectors were obtained from Addgene Company (Watertown, MA, USA).

### 2.2. Cell Culture

Human colorectal cancer cell line DLD-1 (BCRC No. 60132) was purchased from Bioresource Collection and Research Center (BCRC, Hsinchu, Taiwan). DLD-1 cells were cultured in RPMI-1640 medium. The medium was complemented with 10% fetal bovine serum (FBS), 2 mM L-glutamine, 100 units/mL penicillin G, and 100 μg/mL streptomycin. All cells were kept at 37 °C in a 5% CO_2_ incubator. Stock solutions of resveratrol and parecoxib were dissolved in DMSO, and all treated concentrations were adjusted in the culture medium. The concentration of DMSO did not go beyond 0.05%.

### 2.3. MTT Assay for Cell Viability

MTT assay was conducted to measure cell viability. About 4 × 10^4^ cells/well in 0.5 mL of RPMI-1640 medium were cultured in 24-well plates. After growth overnight, DLD-1 cells were incubated with parecoxib (3 μM), resveratrol (50, 100, and 200 μM), and parecoxib (3 μM) combined with resveratrol (50, 100, and 200 μM) for 48 h. The plates were then added with 0.5 mg/mL MTT solution and incubated at 37 °C for another 2 h. The supernatant was separated, and the formazan crystals were dissolved in 1 mL of DMSO. An aliquot of the DMSO lysed solution (200 μL) was obtained from the 24-well plates and transmitted to 96-well reader plates. Optical density (OD) was assessed with a microplate reader (Bio-Rad, Richmond, CA, USA) at 570 nm.

### 2.4. Isobologram Analysis for Synergistic Anti-Cancer Effect

Isobologram analysis [38] was carried out to determine the synergistic anti-cancer effect of a combination of resveratrol and parecoxib. Cell viability after treatment with 0, 1, and 3 μM parecoxib and 0, 100, and 200 μM resveratrol treatment was verified after 48 h, and each concentration was then drawn on each axis of the graph. A diagonal line was plotted between the two concentration spots of each single concentration of resveratrol and parecoxib, denoting the line of additivity as a control. Several values set after treatments with various concentrations of resveratrol and parecoxib in combination were drawn as dots on the graph. The results show antagonism, additivity, or synergy when the dots are localized upon, on or under the diagonal line, respectively.

### 2.5. Overexpression of TXNDC5 in Cancer Cells

The pLAS2w.pPuro and pLAS2w.pPuro-hTXNDC5 vectors were transfected into cells by the X-treme transfection reagent. Approximately 4 × 10^4^ DLD-1 colorectal cancer cells were cultured in a 6-well plate and placed in a 37 °C, 5% CO_2_ incubator for 24 h. After that, 2 mL of serum-free cultured medium was replaced. About 0.2 mL of the serum-free medium was pipetted into the Eppendorf tube, added with 2 μg of vectors and 6 μL of X-tremeGENE™ HP DNA transfection reagent, carefully pipetted and mixed evenly. After reacting at room temperature for 15 min, about 400 μL was obtained and added to the dish. The mixture was shaken evenly, placed back into the incubator at 37 °C for 24 h, and added with 1 μg/mL puromycin antibiotics for 14 days to establish a stable TXNDC5 overexpression cell line. Cells were collected, and their total proteins were extracted to verify the expression level of TXNDC5 by Western blot analyses.

### 2.6. DAPI Staining for Chromatin Condensation and Fragmented Nucleus

DLD-1 colorectal cancer cells (2 × 10^5^ cells/well) were cultured in 6-well plates. After growth overnight, the cells were incubated with parecoxib (3 μM), resveratrol (200 μM), parecoxib (3 μM)/resveratrol (200 μM) combination for 48 h. After drug treatment, the cells were washed with in PBS, fixed with 4% paraformaldehyde for 15 min at room temperature. Afterward, cells were stained with DAPI (1 μg/mL) for 5 min and exposed to three additional PBS washes. The condensed chromatin and fragmented nucleus were detected and photographed under 200× magnification by a fluorescent microscope.

### 2.7. Western Blotting

DLD-1 colorectal cancer cells were planted in 6 cm dishes at a density of 5 × 10^5^ cells/dish for 24 h and then cultured with various drugs as explained in figure legends. Cells were collected as programmed after treatment under various conditions, and total protein concentrations were evaluated by the Bio-Rad protein assay kit. Equivalent total proteins (20–50 μg) of cell lysates were separated through 12% SDS-PAGE and then transferred onto a PVDF membrane for 50–75 min. The PVDF membranes were maintained with 5% nonfat milk in PBST buffer for 1 h to block nonspecific binding. After blocking, the PVDF membranes were incubated with the following primary antibodies at 4 °C overnight: anti-PARP (1:1000), anti-Bax (1:500), anti-Bcl-2 (1:500), anti-GAPDH (1:500), anti-TXNDC5 (1:2500), anti-p53 (1:1000), anti-p-Akt (1:1000), anti-Akt (1:1000), anti-p-JNK (1:500), anti-JNK (1:500), anti-p-p38 (1:500), anti-p38 (1:500), anti-p-ERK (1:500), anti-ERK (1:500) and anti-Tubulin (1:500) antibodies. It was then immersed with secondary antibodies for 1 h at room temperature. The antigen–antibody complexes were evaluated by the enhanced chemiluminescence (Amersham Pharmacia Biotech, Piscataway, NJ, USA) using a chemiluminescence analyzer.

### 2.8. Intracellular ROS Analysis

The level of intracellular ROS was measured by DCFH–DA staining and flow cytometry by. DLD-1 cells were cultured in 6 cm dishes with a density of 4 × 10^5^ cells/dish. Parecoxib (3 μM) alone, resveratrol (200 μM) alone, and parecoxib (3 μM) combined with resveratrol (200 μM) were added, and the cells were treated for 1, 3, and 48 h. After treatment, all cells were incubated with DCFH-DA (10 μM) for intracellular ROS level and determined by using a Backman Coulter cytoFLEX flow cytometer. Cells were treated with 2 mM H_2_O_2_ as the positive control of intracellular ROS. About 10,000 cells were collected and examined per experimental situation via mean fluorescent intensity.

### 2.9. Statistical Analysis

Statistical analysis was conducted using a Student’s *t*-test with SigmaPlot 10.0 software. Data are presented as the mean ± standard deviation from at least three independent experiments. A *p* value < 0.05 was considered statistically significant.

## 3. Results

### 3.1. Parecoxib Enhances Resveratrol to Inhibit Cell Viability in DLD-1 Cells

The non-cytotoxic concentration of parecoxib 3 μM was used to combine with three concentrations of resveratrol and to assess cell viability. As shown in Figure 1, parecoxib combined with resveratrol (200 μM) significantly enhanced the suppression of cell viability compared with resveratrol (200 μM) alone at 48 h treatment. The results showed no significant inhibition in parecoxib combined with 50 and 100 μM resveratrol compared with treatment with 50 and 100 μM of resveratrol alone. However, 1 and 3 μM parecoxib were combined with 100 and 200 μM resveratrol, respectively, resulting in the synergistic effect of reducing cell viability in DLD-1 cells.

### 3.2. Parecoxib Enhances Resveratrol to Induce Apoptosis in DLD-1 Cells

We used DAPI staining to measure nuclear morphological changes and chromatin condensation to demonstrate whether parecoxib can enhance the apoptosis of resveratrol. Parecoxib treatment alone did not cause significant changes in the number of DAPI positive cells in DLD-1 cells. After 48 h of treatment, the resveratrol (200 μM) treatment group had DAPI-positive cells. Combined treatment with 3 μM parecoxib and 200 μM resveratrol significantly increased the numbers of DAPI positive cells (Figure 2A,B). Taken together, these findings demonstrated that, compared with the individual monotherapies, parecoxib and resveratrol combination significantly enlarges the level of apoptosis in DLD-1 colorectal cancer cells.

### 3.3. Parecoxib Enhances Resveratrol to Induce Apoptotic Proteins in DLD-1 Cells

To further explore the apoptotic effect of drugs combination, we examined the expression of apoptotic proteins including cleaved PARP, p53, Bax, and Bcl-2 after 48 h treatments. As evaluated by Western blot, the cleaved PARP was markedly enhanced when DLD-1 cells were treated with parecoxib and resveratrol combination compared with resveratrol treatment alone (Figure 3). We next assessed the apoptotic pathway provoked by the combined treatment. Slightly increased levels were detected in the expression of protein p53 and Bax in DLD-1 cells treated with parecoxib alone. Treatment with resveratrol alone resulted in a distinctly increased expression of p53 and Bax and decreased Bcl-2 expression. However, the expression of p53 and Bax noticeably further prominently increased when cells were treated with parecoxib and resveratrol combination, as compared with those treated with each drug alone.

### 3.4. Role of the PI3K/Akt Signaling Pathway in the Combination Effects of Parecoxib and Resveratrol

To evaluate whether the PI3K/Akt signaling pathway was included in the synergistic effects of parecoxib and resveratrol, the expression of Akt phosphorylation in DLD-1 cells was assessed by Western blotting. As shown in Figure 4, the phosphorylation of Akt was diminished in cells treated with resveratrol and there was no obvious change after parecoxib treatment. However, the phosphorylation of Akt was distinctly lower in cells exposed to combination treatment when compared with resveratrol alone. Hence, parecoxib may exhibit the enhanced repression of resveratrol on the cell viability of colorectal cancer cells by suppressing the PI3K/Akt signaling pathway.

### 3.5. Role of TXNDC5 in the Combination Effects of Parecoxib and Resveratrol

To evaluate the role of TXNDC5 in the combination effects of parecoxib and resveratrol, the expression of TXNDC5 in DLD-1 cells were assessed by Western blotting. As shown in Figure 5A, the expression of TXNDC5 did not obviously change between cells treated with parecoxib alone and resveratrol alone. However, the expression of TXNDC5 was reduced to a greater extent after parecoxib and resveratrol combination treatment. Next, we evaluated whether TXNDC5 plays a critical role on the anti-cancer effect in drug combinations; the DLD-1 cells were transfected with an empty vector or TXNDC5 vector, and then we evaluated the expression of TXNDC5 by Western blot. As shown in Figure 5B, the expression of TXNDC5 was exhibited to a distinctly greater extent in TXNDC5 plasmid-transfected cells compared with empty plasmid-transfected cells. After treatment with parecoxib and resveratrol combination, the cell viability in TXNDC5 overexpressed DLD-1 cells was significantly increased compared with empty plasmid-transfected DLD-1 cells (Figure 5C). These results imply that TXNDC5 inhibition is a key mechanism in the anti-cancer effect of parecoxib and resveratrol combination.

### 3.6. Apoptotic Role of TXNDC5 in the Combination Effects of Parecoxib and Resveratrol

We further evaluated the apoptotic role of TXNDC5 in the anti-cancer effect of parecoxib and resveratrol, the TXNDC5 vector-transfected DLD-1 cells and empty vector-transfected cells were treated with parecoxib and resveratrol combination, and then we assessed the expression of cleaved PARP and Bax by Western blotting. As shown in Figure 6, the expression of cleaved PARP and Bax was increased to 2.05-fold and 1.32-fold in empty plasmid-transfected cells treated with parecoxib and resveratrol combination compared with untreated empty plasmid-transfected cells. However, the expression of cleaved PARP and Bax decreased to 0.96-fold and 1.10-fold in TXNDC5 plasmid-transfected cells after parecoxib and resveratrol combination compared with untreated TXNDC5 plasmid-transfected cells. These results suggest that TXNDC5 inhibition is involved in the apoptotic event in the anti-cancer effect of parecoxib and resveratrol combination.

### 3.7. Role of MAPK Signaling in the Combination Effects of Parecoxib and Resveratrol

The MAPK pathway is an important signaling pathway in the regulation of apoptosis. We further explored the expression of p-JNK, p-p38, and p-ERK by Western blot in DLD-1 cells treated with parecoxib alone, resveratrol alone, and parecoxib and resveratrol combination. As shown in Figure 7A, p-JNK expression was slightly increased in parecoxib alone and resveratrol alone treatment. Notably, parecoxib and resveratrol combination induced a large amount of p-JNK expression compared with resveratrol alone. The p-p38 expression was slightly increased in parecoxib alone. In resveratrol alone and drug combination treatment, the p-p38 expression was increased by more than 2.5-fold. The p-p38 expression in parecoxib and resveratrol combination is more than that in resveratrol alone. The p-ERK expression was increased more than 2.5-fold in parecoxib alone compared with the untreated group. Treatment with resveratrol alone and parecoxib and resveratrol combination resulted in a more than 5.5-fold increase in p-ERK expression. However, the p-ERK expression in the drug combination is slightly lower than treatment with resveratrol alone. To evaluate whether JNK and p38 signaling are included in the enhancement of cell viability inhibition and apoptosis of parecoxib and resveratrol combination, SP600125 (a JNK inhibitor) and SB203580 (a p38 inhibitor) were pretreated for 1 h, then treated with parecoxib and resveratrol combination for 48 h. After drug treatment, the cell viability and the expression of cleaved PARP and Bax were assessed by MTT assay and Western blot, respectively. As shown in Figure 7B,C, pretreatment of JNK and p38 inhibitors resulted in significantly recovered cell viability and a lower level of the expression of cleaved PARP and Bax compared with parecoxib and resveratrol combination. These results imply that p38 and JNK signaling participated in the enhancement of cell viability inhibition and apoptosis of parecoxib and resveratrol combination.

### 3.8. Effects of Parecoxib and Resveratrol on Intracellular ROS in DLD-1 Cells

ROS status is an important factor in cell proliferation. We evaluate whether parecoxib and resveratrol treatment affected the ROS status in DLD-1 cells. As shown in Figure 8A, there was no obvious change in intracellular ROS after treatment with parecoxib and resveratrol for 1 h. The intracellular ROS are significantly decreased to 55% and 60% after 3 h of resveratrol alone and combined treatment, respectively, compared to untreated cells. After 48 h of treatment, the intracellular ROS treated with resveratrol alone and combined treatment decreased below 50% (Figure 8B). There is no difference between resveratrol alone and the combination.

## 4. Discussion

In Figure 1, the lowest concentration (50 µM) of resveratrol is already quite active in cell viability inhibition. However, the lowest concentration (50 µM) of resveratrol combined with a concentration (3 µM) of parecoxib could not exhibit more inhibition of cell viability compared with treatment with resveratrol (50 µM) alone. In contrast, the highest concentration of resveratrol (200 µM) combined with parecoxib (3 µM) appeared to significantly inhibit cell viability compared with treatment with resveratrol (200 µM) alone. For this reason, we used the highest concentration (200 µM) of resveratrol to combine with parecoxib (3 µM) in all subsequent experiments.

TXNDC5 abnormally appeared in several cancers, such as colorectal cancer [32]. Presently, TXNDC5 is thought of as a cancer-enhancing gene [39]. It can stimulate cell proliferation, promote cancer growth, suppress apoptosis, defend cells from oxidative stress, and accelerate the development of disease. Moreover, targeting TXNDC5 in the therapy of diseases has exhibited favorable treatment potentials. TXNDC5 can be exercised as a therapeutic target for cancers. Inhibition of TXNDC5 expression in various cancers such as gastric cancer [39], laryngeal squamous cell carcinoma [40], non-small cell lung carcinoma [41], pancreatic cancer [42], cervical cancer [31], liver cancer [43], and castration-resistant prostate cancer [44] can provoke cell apoptosis and reduce cell proliferation and migration. These results recommend that TXNDC5 can be used as a therapeutic target for cancers. Cetuximab could diminish TXNDC5 expression, thereby augmenting the generation of ROS and developing the endoplasmic reticulum stress-related apoptosis of laryngeal squamous cell carcinoma cells [40]. In addition, knockdown of TXNDC5 can result in clear cell renal cell carcinoma cells sensitive to chemotherapy drugs such as 5-fluorouracil and camptothecin and suppress the growth, migration, and invasion of clear cell renal cell carcinoma cells [45].

The anti-cancer action of resveratrol is to stimulate apoptosis in cancer cells, mediated by the protein p53. Resveratrol can affect the condition of cancer cells through the p53 pathway by enlarging the anti-colorectal cancer force of p53 [46]. In agreement with previous studies validating that resveratrol prompts p53-related apoptosis, resveratrol stimulated p53-related apoptosis in DLD-1 cells in the current study, as verified by the raised levels of PARP cleavage and Bax. TXNDC5 is an endoplasmic reticulum (ER) protein protective against ER stress-associated apoptosis [40]. However, our results showed that resveratrol did not increase TXNDC5 expression in DLD-1 cells; this may be due to the loss of a protective event of DLD-1 cells to provoke apoptosis under resveratrol treatment. The explicit effect of parecoxib was to enhance resveratrol-produced apoptosis, which was demonstrated by the following: (i) the significant augmentation of the cell viability inhibition and the apoptotic effect of resveratrol combined with parecoxib; (ii) parecoxib and resveratrol combination obstructed the expression of TXNDC5, and obviously augmented the apoptotic effect of resveratrol; and (iii) the inhibition of cell viability in the drug combination was recovered by overexpressing TXNDC5. These results suggest that TXNDC5 inhibition is a critical mechanism underlying the apoptotic effect of parecoxib and resveratrol combination in DLD-1 cells.

An important substrate of Akt that induces the mitochondrial apoptotic pathway is caspase-9, which is incapacitated by Akt by the phosphorylation at Ser196 [47]. Caspase-9 deactivation subsequently results in the deactivation of caspase-3 and repression of caspase-dependent apoptosis [48]. Moreover, stimulation of the Akt signaling pathway downregulates the expression of p53 and Bax. Consequently, Akt stimulates cell cycle progression, avoids apoptosis and enhances cell proliferation [49]. Resveratrol reduces the function of Akt and its downstream targets, hence stimulating apoptosis and cell cycle arrest, together with repressing cell proliferation in colon cancer cells [49]. Resveratrol enhances the sensitivity of ovarian cancer cells to cisplatin, causing them to be more sensitive to apoptotic cell killing. The effect of resveratrol was confirmed to initiate from its capacity to trigger p38 MAPK in particular, and diminish Akt activation [37]. Consistent with previous findings, our results found that resveratrol treatment alone diminished p-Akt expression and increased the expression of cleaved PARP, p53, and Bax. Although treatment with parecoxib alone did not affect the expression of p-Akt, it moderately increased the expression of p53 and slightly increased the expression of cleaved PARP and Bax. Parecoxib and resveratrol combination can better inhibit p-Akt expression and increase the expression of cleaved PARP, p53, and Bax compared with resveratrol alone. These results illustrate that parecoxib can enhance resveratrol-induced apoptosis by inhibiting the p-Akt signaling pathway, accompanied by increasing apoptotic-related proteins expression in colorectal cancer calls.

Recently, our study demonstrated that parecoxib can synergistically enhance 5-fluorouracil to repress metastasis in human colorectal cancer [28]. Resveratrol can moderate many cellular pathways correlated with tumorigenesis but it is less effective in colon cancer [50]. In human chronic myelogenous leukemia cells, resveratrol induces apoptosis via activating two MAPK family members, p38 and JNK, and preventing the activation of another MAPK family member, ERK [51]. We postulated that (1) parecoxib may enhance the anti-cancer effect of resveratrol in human colorectal cancer; (2) and resveratrol may employ like signaling pathways to disturb human colorectal cancer cells. Our results from the Western blot evaluation distinctly showed that resveratrol marked the induced phosphorylation of ERK and p38. However, the phosphorylation of JNK only slightly increased in treatment with resveratrol alone. When resveratrol was combined with parecoxib, the phosphorylation of p38 was further slightly increased. We also detected the activation of the ERK upon drug combination but detected no stronger enhancement in the phosphorylation level of the ERK kinase. Parecoxib and resveratrol combination enhanced the phosphorylation of JNK compared with resveratrol alone treatment. Moreover, SP600125 (a JNK inhibitor) and SB203580 (a p38 inhibitor) could alleviate the apoptosis and caused a rebound in the inhibition of cell viability caused by parecoxib and resveratrol combination. These results imply that parecoxib may particularly control the JNK and p38 signal transduction pathways to enhance the anti-cancer effect of resveratrol in human colorectal cancer. Parecoxib could enhance the effectiveness of resveratrol-based chemotherapy against human colorectal cancer. Celecoxib, a COX-2 inhibitor, can enhance apoptosis via upregulating the p-JNK and p-p38 pathway in liver cancer cells [52]. The enhancement of JNK activity under the influence of various anti-cancer compounds provokes the apoptosis of various human cancer cells [53,54,55]. Resveratrol was described to provoke the activation of JNK, which resulted in induced CHOP-related apoptosis in human colon cancer [56]. Our results indicate the augmented cytotoxic effects of parecoxib on colorectal cancer cell upon resveratrol treatment by the forceful activation of JNK and p38 pathways; as such, colorectal cancer cells become more sensitive to apoptotic cell death stimulation.

Hypoxia causes TXNDC5 expression by increasing hypoxia inducible factor-1α in vivo, thereby suppressing hypoxia-stimulated ROS/ER stress signaling and elevating the reproduction and survival of colorectal cancer cells [57]. These results indicate that there is some special relationship between TXNDC5 and intracellular ROS. Our present study shows that treating cells with resveratrol alone for 3 h and 48 h resulted in reducing intracellular ROS. This result indicates that the antioxidant property of resveratrol leads to the reduction in intracellular ROS. However, parecoxib treatment alone did not change the intracellular ROS levels in DLD-1 cells. Even if parecoxib and resveratrol combination cannot reduce intracellular ROS compared with resveratrol alone treatment, this result shows that the reduction in ROS in cells treated with combined drugs also comes from the antioxidant property of resveratrol. However, treatment with resveratrol alone did not diminish the expression of TXNDC5. Although the combined drugs treatment reduced TXNDC, the amount of intracellular ROS did not decrease. These results illustrate that the reduction in TXNDC5 in the drug combination treatment is not directly related to the change in intracellular ROS in DLD-1 colorectal cancer cells.

Resveratrol can target steroid receptors signaling and result in a potential anti-cancer effect in the treatment of hormone-dependent cancer [58]. We speculate that resveratrol may inhibit cell viability through targeting some unknown receptors in DLD-1 cells. In Figure 1A, the cell viability of cells treated with 100 µM of resveratrol alone is slightly higher than 50 µM of resveratrol alone. It is possible that the concentration of 50 µM resveratrol had reached a saturated state by binding to some unknown receptors, causing the greatest cell viability suppression, while the concentrations of 100 and 200 µM resveratrol might produce a competitive effect with receptors, resulted in an effect of inhibiting cell viability that is slightly less than the resveratrol of 50 µM. This speculation must be further studied in the future.

## 5. Conclusions

Parecoxib augments the sensitivity of colorectal cancer cells to resveratrol, causing them to become more sensitive to cell viability inhibition and apoptosis. Parecoxib provides a unique key which can not only enhance resveratrol in reducing cancer cell viability, but also enhance apoptosis via the suppression of Akt activation and TXNDC5 expression, upregulating the JNK and p38 MAPK signaling pathway, as well as increasing p53 and Bax expression and decreasing Bcl-2 expression in colorectal cancer, thereby enlarging cleaved PARP and enhancing apoptosis (Figure 9). The significant limitations of this study are that it is only an in vitro study, and that many various body, diet, and drug-dependent factors may have had a significant influence on such interactions. This study only has preliminary significance, showing that parecoxib and resveratrol combined treatment may offer a hopeful future for colorectal cancer patients. Much more research (especially in vivo studies) is needed to fully elucidate the importance of combining parecoxib with resveratrol in the future.

## Figures and Tables

**Figure 1 nutrients-16-03020-f001:**
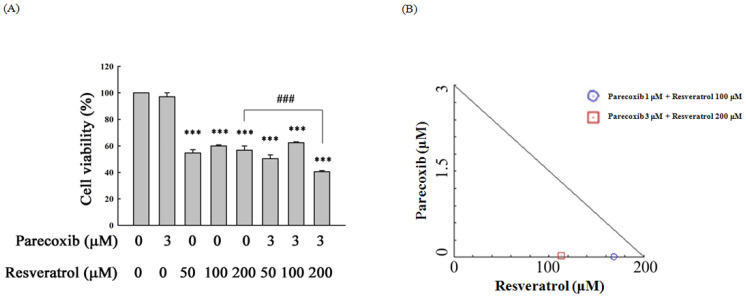
(**A**) Cell viability and (**B**) isobologram analysis in parecoxib and resveratrol treatment at 48 h. After drug treatment, cell viability was measured by MTT analysis. In isobologram analysis, the points under the backslash line show the synergistic effect. Significant differences in the untreated group (UN) and resveratrol are displayed as follows: *p* < 0.001 (***) and *p* < 0.001 (###), respectively.

**Figure 2 nutrients-16-03020-f002:**
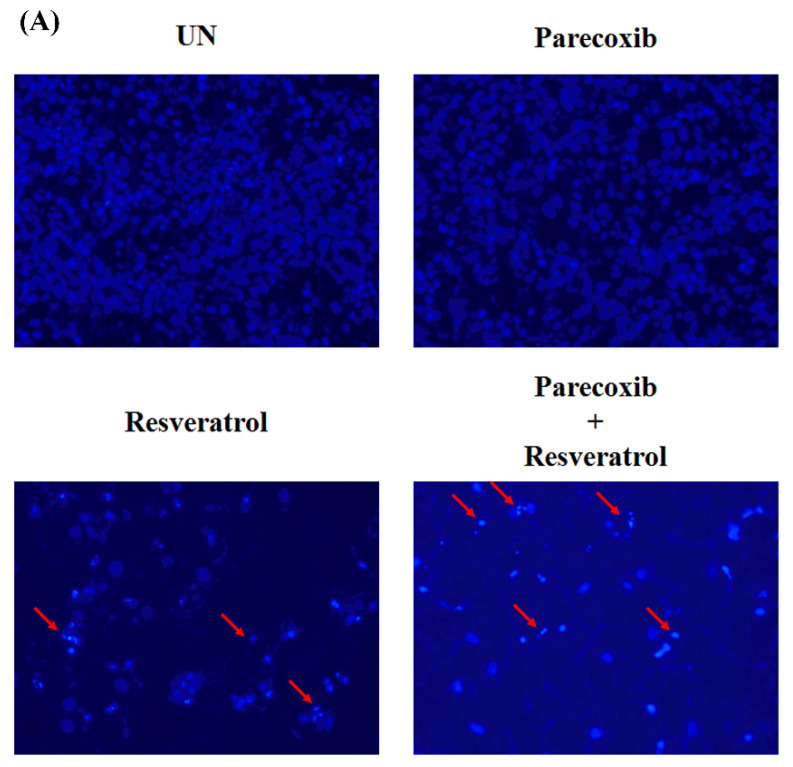
Effect of parecoxib and resveratrol on apoptotic morphology and apoptosis. (**A**) After treatment, chromatin condensation and fragmented nucleus were evaluated using DAPI staining and monitored with fluorescence microscopy (magnification 200×). Red arrows indicate the DAPI positive cells. (**B**) Number of DAPI-positive apoptotic cells per slide was calculated by counting apoptotic cells in five different fields. Each value indicates a mean ± SD (*n* = 5). Significant difference in the resveratrol is displayed as follows: *p* < 0.01 (**).

**Figure 3 nutrients-16-03020-f003:**
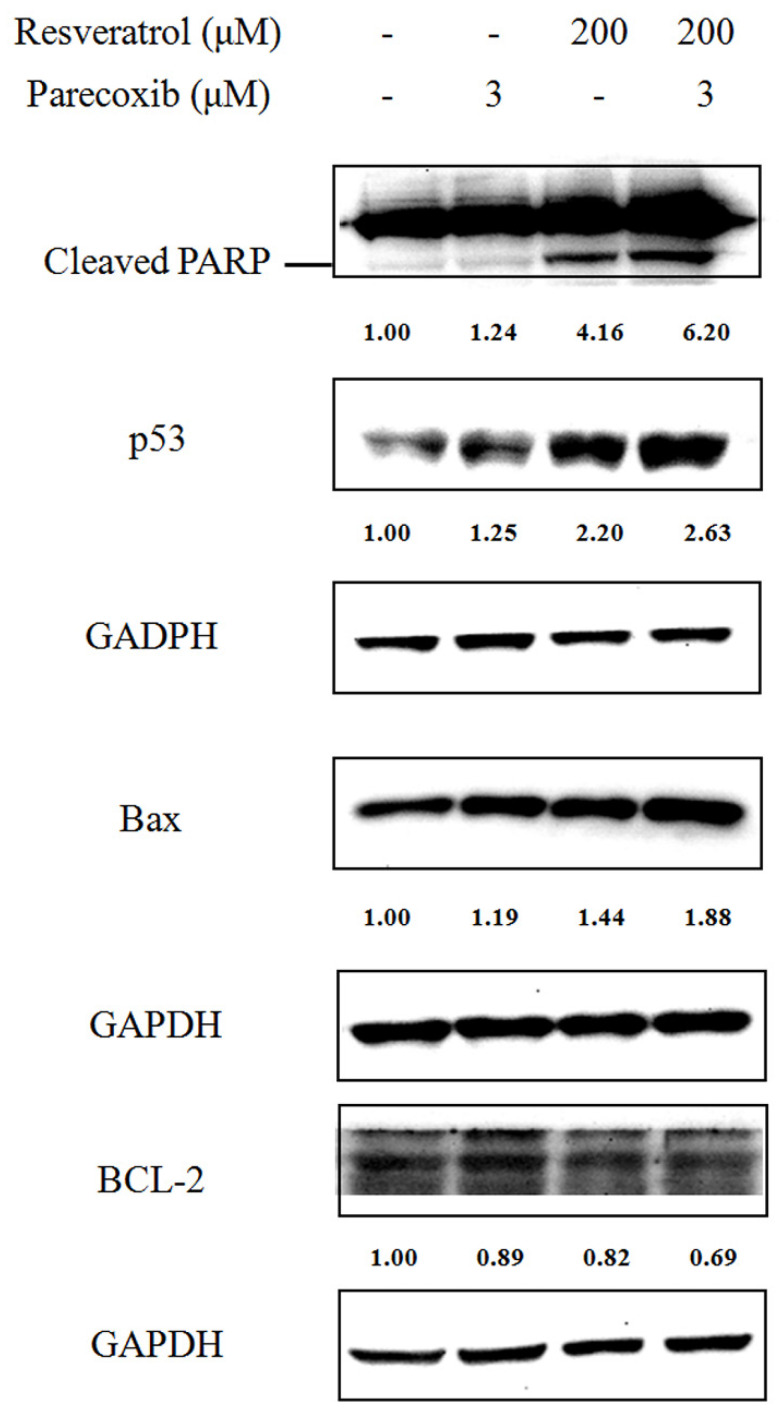
Effect of parecoxib and resveratrol on apoptotic proteins. After treatment, the levels of protein expression were measured using the extracted proteins and determined by Western blot. GAPDH were used as internal control.

**Figure 4 nutrients-16-03020-f004:**
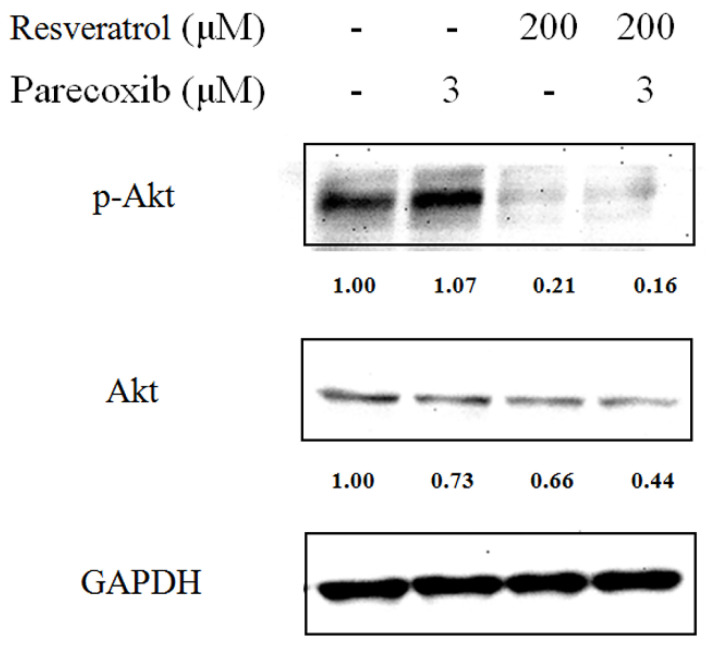
Effect of parecoxib and resveratrol on the expression of p-Akt and Akt. After treatment, the levels of protein expression were measured using the extracted proteins and determined by Western blot. GAPDH were used as internal control.

**Figure 5 nutrients-16-03020-f005:**
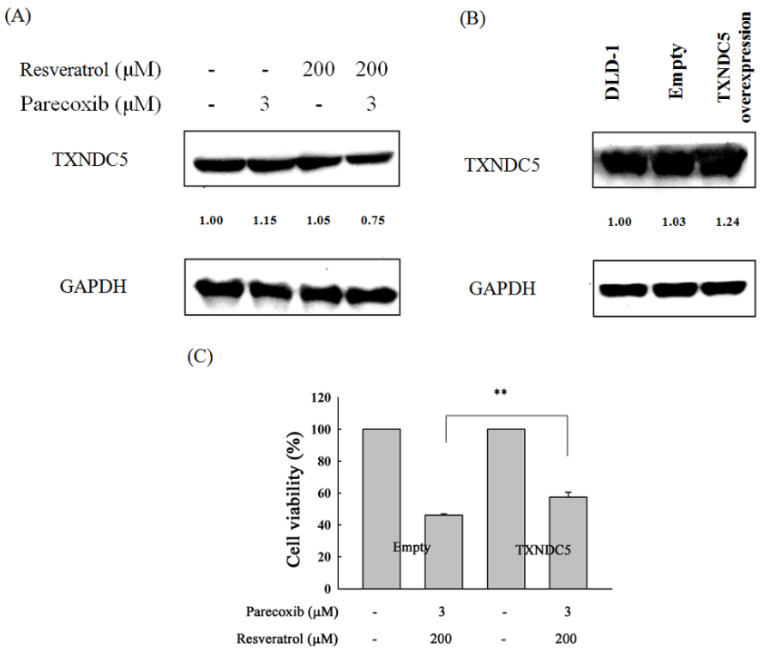
Effect of TXNDC5 overexpression in parecoxib- and resveratrol-treated DLD-1 cells. (**A**,**B**) Expression of TXNDC5 was measured by Western blot. GAPDH were chosen as loading control. (**C**) The cell viability was measured by MTT assay. These experiments were performed at least three times, and a representative experiment is presented. Data are shown as the mean ± SD of separate tests. Significant differences are expressed as *p* < 0.01 (**).

**Figure 6 nutrients-16-03020-f006:**
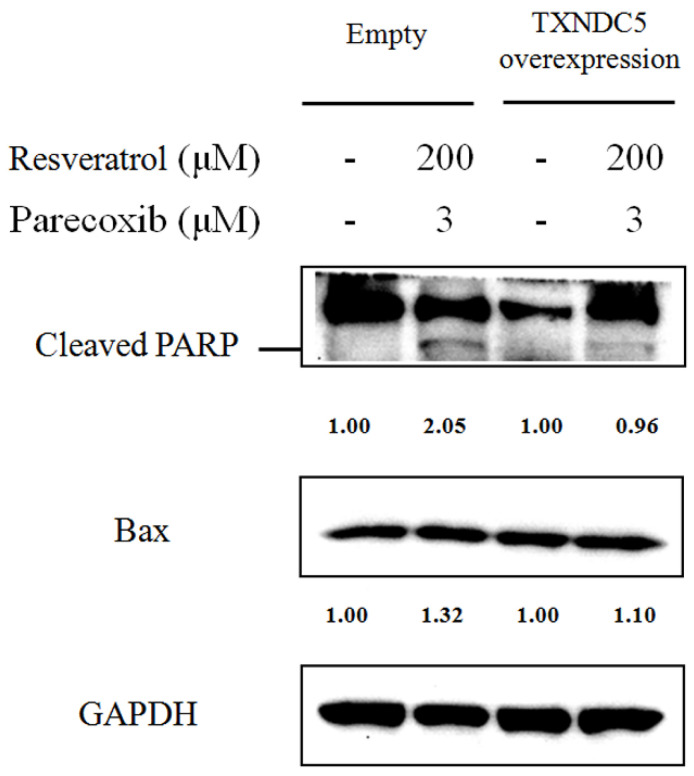
Role of TXNDC5 on apoptosis in parecoxib and resveratrol combination. After treatment, the levels of protein expression were measured using the extracted proteins and determined by Western blot. GAPDH were used as internal control.

**Figure 7 nutrients-16-03020-f007:**
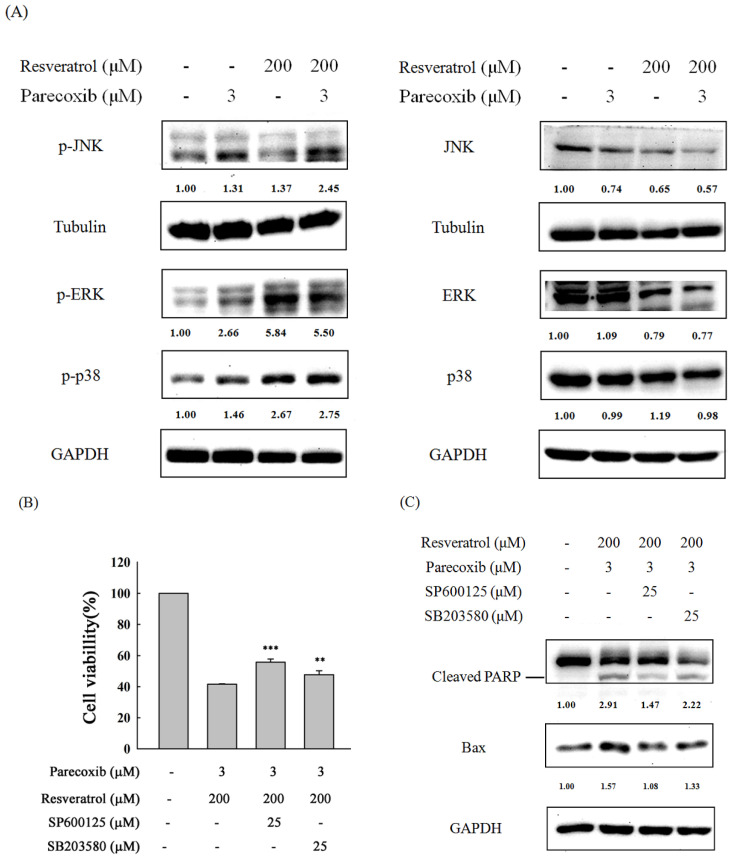
Effect of MAPKs signaling on apoptosis in parecoxib and resveratrol combination. (**A**) Phosphorylation and non-phosphorylation of MAPKs were measured by Western blot. GAPDH and tubulin were selected as loading control. (**B**) The cell viability was analyzed by MTT assay. These experiments were performed at least three times, and a representative experiment is presented. Data are shown as the mean ± SD of separate tests. Significant differences are expressed as *p* < 0.01 (**) and *p* < 0.001 (***). (**C**) Cleaved PARP and Bax were detected by Western blot. GAPDH were selected as loading control.

**Figure 8 nutrients-16-03020-f008:**
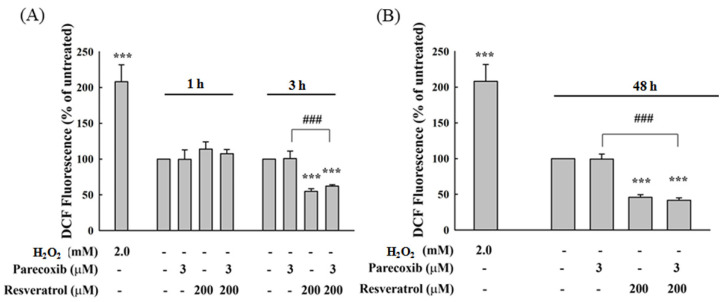
Effect of parecoxib and resveratrol on intracellular ROS in DLD-1 cells. (**A**) Treatment of 1 and 3 h; (**B**) 48 h treatment. After treatment, all cells were stained with DCFH-DA for intracellular ROS detection and determined by a flow cytometer. H_2_O_2_ (2.0 mM) treatment was selected as an intracellular ROS positive control. The data are shown as the mean ± SD (*n* = 5–8) of individual experiments. Significant differences in the untreated group (UN) and parecoxib are shown as follows: *p* < 0.001 (***) and *p* < 0.001 (###).

**Figure 9 nutrients-16-03020-f009:**
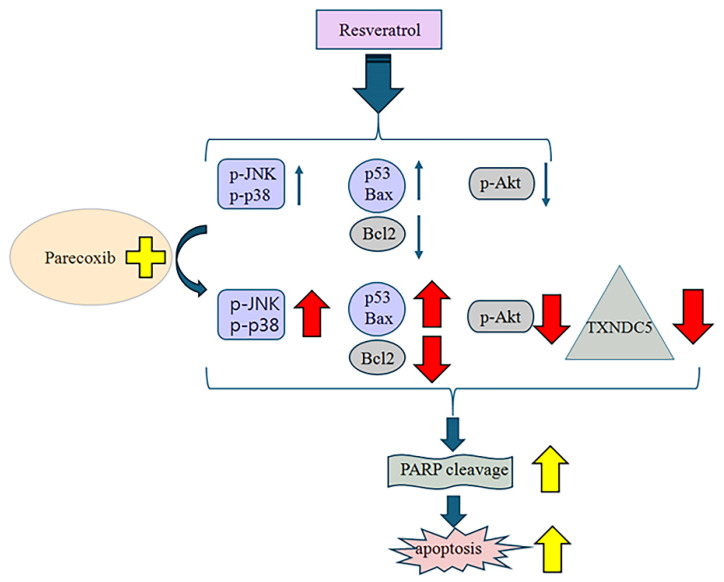
Proposed model of parecoxib and resveratrol combination that enhanced apoptosis in colorectal DLD-1 cancer cells.

## Data Availability

The data that support the findings of this study are contained within the article.

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
