# Peer review of "Parecoxib Enhances Resveratrol against Human Colorectal Cancer Cells through Akt and TXNDC5 Inhibition and MAPK Regulation"

_nutrients, 2024, doi:10.3390/nu16173020_

Round 1

Reviewer 1 Report

Comments and Suggestions for Authors

The manuscript describes the effect of the COX-2 inhibitor parecoxib on the activity of the well-known dietary polyphenol resveratrol against colon cancer cells. The authors described interesting mechanisms of action for this promising combination therapy. However, the manuscript has some flaws, and I recommend major revision:

Section 3.1., Figure 1A: The cell viability of cells treated the combination of 3 µM parecoxib with 100 µM resveratrol is higher than in cells treated with 3 µM parecoxib and 50 µM resveratrol. Maybe the authors can explain and discuss this strange result. In addition, the incubation time (48 h?) should be mentioned in the caption or in the associated main text of 3.1..

Section 3.2.: Is it possible to quantify the numbers of DAPI-positive cells?

Figures 1, 8, and 9, caption: The font size of ´´resveratrol´´ differs from the other text of the caption. Please correct.

Line 405: The quotation ´´5-fluorescence´´ needs to be explained or corrected.

The manuscript needs some English correction (e.g., ´´were pretreatment for 1 h´´, ´´suppression is involves in´´, ´´along with increased the expression´´, ´´in our present study finds that´´, etc.).

Reference 48: Please abbreviate the journal name properly.

Comments on the Quality of English Language

n.a.

Author Response

Response to Reviewer 1:

We sincerely thank you for your constructive and valuable comments that helped us improve our manuscript. Herein, we provide our point-by-point responses to the comments along with a description of all the changes that have been made and highlighted in blue in the revised manuscript.

The manuscript describes the effect of the COX-2 inhibitor parecoxib on the activity of the well-known dietary polyphenol resveratrol against colon cancer cells. The authors described interesting mechanisms of action for this promising combination therapy. However, the manuscript has some flaws, and I recommend major revision:

Section 3.1., Figure 1A: The cell viability of cells treated the combination of 3 µM parecoxib with 100 µM resveratrol is higher than in cells treated with 3 µM parecoxib and 50 µM resveratrol. Maybe the authors can explain and discuss this strange result. In addition, the incubation time (48 h?) should be mentioned in the caption or in the associated main text of 3.1..

Resveratrol can target on steroid receptors signaling and result in potential anti-cancer effect in the treatment of hormone-dependent cancer [62]. We speculate that resveratrol may inhibit cell viability through targeting some unknown receptors in DLD-1 cells. In Figure 1A, the cell viability of cells treated 100 µM resveratrol alone is slightly higher than 50 µM resveratrol alone. It is possible that the concentration of 50 µM resveratrol had reached a saturated state by binding to some unknown receptors causing the greatest cell viability suppression, while the concentrations of 100 and 200 µM resveratrol might produce a competitive effect with receptors, resulted in an effect of inhibiting cell viability that is slightly less than the resveratrol of 50 µM. This speculation must be further studied in the future. (Please see page 14, the last paragraph)

We added the incubation time (48 h) in the caption and in the associated main text of 3.1. (Please see page 5, line 248 and Figure 1 caption, page 6, line 255)

Section 3.2.: Is it possible to quantify the numbers of DAPI-positive cells?

We added the Figure 2B to show the quantitation of the numbers of DAPI-positive cells. (Please see page 7, Figure 2B)

Figures 1, 8, and 9, caption: The font size of ´´resveratrol´´ differs from the other text of the caption. Please correct.

We have collected the font size of ´´resveratrol´´. (Please see Figure 1 caption, page 6, line 254; Figure 8 caption, page 12, line 390; Figure 9 caption, page 15, line 511)

Line 405: The quotation ´´5-fluorescence´´ needs to be explained or corrected.

The ´´5-fluorescence´´ is typo. We have revised to ´´5-fluorouracil´´. (Please see page 13, lines 457-458)

The manuscript needs some English correction (e.g., ´´were pretreatment for 1 h´´, ´´suppression is involves in´´, ´´along with increased the expression´´, ´´in our present study finds that´´, etc.).

We have corrected these English. (Please see page 11, line 363; page 10, line 341; page 13, line 449; page 14, lines 488-489)

Reference 48: Please abbreviate the journal name properly.

We have abbreviated the journal name to “Eur J Med Res”. (Please see page 18, Reference 61)

Reviewer 2 Report

Comments and Suggestions for Authors

In this study, authors have studied the mechanisms by which parecoxib enhanced the effect of resveratrol against human colorectal cancer DLD-1 cells.

By means of appropriate methods they found that this combination enhanced apoptosis by inhibiting the expression of thioredoxin domain containing 5 (TXNDC5) and Akt phosphorylation and the involvement of JNK/p38 MAPK pathways.

Thus, the results seem to support the conclusions that parecoxib may be combined with resveratrol to increase its efficacy against colorectal cancer cells.

However, some points need to be better clarified to improve the understanding of the proposed data.

-In Fig.1 it is evident that resveratrol is already quite active at the lowest concentration (50 µM); however, in all subsequent experiments it was used at the highest concentration of the 3 tested (200 µM). Please explain better the reason for this choice.

-In section 2.2. Cell Culture, it is reported that human colorectal cancer cell line DLD-1 was cultured in RPMI-1640 medium. So why is DMEM medium also reported in the 2.1. Reagents and Chemicals section? Were the cells perhaps maintained, grown in different media depending on the experiment? in this case, explain why.

-For the Isobologram Analysis for Synergestic Anticancer Effect of the two compounds no reference is reported. Given the importance of this analysis it is better to indicate at least one important bibliographical reference.

-In Fig. 5C the significance is reported even if graphically it is not so evident, so please indicate at least the number of experiments performed. The same for Fig. 7.

Comments on the Quality of English Language

some editorial oversights.

Author Response

Response to Reviewer 2:

We sincerely thank you for your constructive and valuable comments that helped us improve our manuscript. Herein, we provide our point-by-point responses to the comments along with a description of all the changes that have been made and highlighted in blue in the revised manuscript.

In this study, authors have studied the mechanisms by which parecoxib enhanced the effect of resveratrol against human colorectal cancer DLD-1 cells.

By means of appropriate methods they found that this combination enhanced apoptosis by inhibiting the expression of thioredoxin domain containing 5 (TXNDC5) and Akt phosphorylation and the involvement of JNK/p38 MAPK pathways.

Thus, the results seem to support the conclusions that parecoxib may be combined with resveratrol to increase its efficacy against colorectal cancer cells.

However, some points need to be better clarified to improve the understanding of the proposed data.

-In Fig.1 it is evident that resveratrol is already quite active at the lowest concentration (50 µM); however, in all subsequent experiments it was used at the highest concentration of the 3 tested (200 µM). Please explain better the reason for this choice.

In Figure 1, the lowest concentration (50 µM) of resveratrol is already quite active in cell viability inhibition. However, the lowest concentration (50 µM) of resveratrol combined with concentration (3 µM) of parecoxib could not exhibit more inhibition of cell viability compared with resveratrol (50 µM) alone treatment. In contrast, the highest concentration of resveratrol (200 µM) combined with parecoxib (3 µM) appeared significantly inhibition of cell viability compared with resveratrol (200 µM) alone treatment. For the reason, we used the highest concentration (200 µM) of resveratrol to combine with parecoxib (3 µM) in all subsequent experiments. (Please see Discussion section, page 12, the first paragraph)

-In section 2.2. Cell Culture, it is reported that human colorectal cancer cell line DLD-1 was cultured in RPMI-1640 medium. So why is DMEM medium also reported in the 2.1. Reagents and Chemicals section? Were the cells perhaps maintained, grown in different media depending on the experiment? in this case, explain why.

Thank you for your detailed censor. The human colorectal cancer cell line DLD-1 was cultured in RPMI-1640 medium. We have deleted the DMEM in the 2.1. Reagents and Chemicals section. (Please see 2.1. Reagents and Chemicals section, page 4, line 154)

-For the Isobologram Analysis for Synergestic Anticancer Effect of the two compounds no reference is reported. Given the importance of this analysis it is better to indicate at least one important bibliographical reference.

We have added one bibliographical reference to our manuscript. (Please see 2.4. Isobologram Analysis for Synergestic Anticancer Effect section, page 4, line 185 and page 17, reference 39, lines 629-630)

-In Fig. 5C the significance is reported even if graphically it is not so evident, so please indicate at least the number of experiments performed. The same for Fig. 7.

We have indicated at least the number of experiments performed in Fig 5C and Fig 7. These experiments were performed at least three times, and a representative experiment is presented. (Please see Figure 5C legend, page 10, lines 327-328 and Figure 7B legend, page 11, lines 375-376)

some editorial oversights.

We have corrected the editorial oversights.

Reviewer 3 Report

Comments and Suggestions for Authors

The paper is interesting and may be suitable for the Journal, but first several improvements must be performed:

1. lines 37-43 - the initial paragraph was to generally written. The Authors should describe in more details, that phytochemicals are "non-nutrients" - compounds which are not essential to the organism, but may play a very beneficial role to the body, including prevention of NCDs (Non-communicable diseases), like CVD, cancer, diabetes etc. I recommend to strongly improve this part. Please provide also significant dietary sources of resveratrol, amounts in which it is consumed with the normal, human diet. Some data on its absorption and metabolism are also crucial, since the study is in vitro only and metabolism of this compound in the body may have significant effect on its activity and interaction with parecoxib. 

2. line 41 - "scientists are busy" - this is a colloquialism. It shouldn't be written like that in an official scientific journal. Please correct it

3. line 46 - "resveratrol is generated". It is rather produced by plants. It is a plant secondary meatoblite. 

4. lines 46-50 - since the authors write such a detailed history of resveratrol, provide the date of its discovery, they should first of all write that this compound occurs in the form of two geometric isomers - trans and cis and that only the former shows biological activity.

5. line 64 - "Parecoxib is a generally consumed" - food is consumed, drugs are rather taken or prescribed. Please correct. 

6. lines 64-65 - please provide more details on parecoxib. Why inhibition of COX-2 is so important in comparison to the first generation drugs, which mainly inhibit COX-1. Please also indicate which serious side effects may COX-2 inhibitors provide. 

7. lines 118 - was it trans-resveratrol?

8. The research is interesting, but the conclusions are to general. The Authors sholud write significant limitations of the study. It is only in vitro study and many various body, diet and drug-dependent factors may have significant influence on such interactions. The Authors should clearly described, that this study has only preliminary significance and much more research (especially in vivo studies) are needed to fully elucidate importance of combining parecoxib with resveratrol. 

Author Response

Response to Reviewer 3:

We sincerely thank you for your constructive and valuable comments that helped us improve our manuscript. Herein, we provide our point-by-point responses to the comments along with a description of all the changes that have been made and highlighted in blue in the revised manuscript.

The paper is interesting and may be suitable for the Journal, but first several improvements must be performed:

  1. lines 37-43 - the initial paragraph was to generally written. The Authors should describe in more details, that phytochemicals are "non-nutrients" - compounds which are not essential to the organism, but may play a very beneficial role to the body, including prevention of NCDs (Non-communicable diseases), like CVD, cancer, diabetes etc. I recommend to strongly improve this part. Please provide also significant dietary sources of resveratrol, amounts in which it is consumed with the normal, human diet. Some data on its absorption and metabolism are also crucial, since the study is in vitro only and metabolism of this compound in the body may have significant effect on its activity and interaction with parecoxib. 

Thank you for your suggestions. We have improved the first paragraph according to your comments. (Please see Introduction section, the first paragraph, page 1)

We also provide the significant dietary sources of resveratrol, amounts in which it is consumed with the normal, human diet and the data about the absorption and metabolism of resveratrol in the second paragraph. (Please see page 2, the first paragraph, lines 46-65)

  1. line 41 - "scientists are busy" - this is a colloquialism. It shouldn't be written like that in an official scientific journal. Please correct it

We correct the sentence to “scientists are working to uncover the functions of dietary phytochemicals in plant foods to provide evidence for cancer prevention.”. (Please see Introduction section, the first paragraph, page 1, lines 43-44)

  1. line 46 - "resveratrol is generated". It is rather produced by plants. It is a plant secondary meatoblite. 

We revised the sentence to “…resveratrol is a secondary metabolite derived from plant resistance to pathogenic attack and environmental stress.”. (Please see page 2, the first paragraph, lines 67-68)

  1. lines 46-50 - since the authors write such a detailed history of resveratrol, provide the date of its discovery, they should first of all write that this compound occurs in the form of two geometric isomers - trans and cis and that only the former shows biological activity.

We added the sentence “Resveratrol occurs in the form of two geometric isomers - trans and cis and that only the trans geometric isomers shows biological activity.”. (Please see page 2, line 65-67)

  1. line 64 - "Parecoxib is a generally consumed" - food is consumed, drugs are rather taken or prescribed. Please correct. 

We have revised "consumed" to "prescribed" in the sentence. (Please see page 2, lines 96-97)

  1. lines 64-65 - please provide more details on parecoxib. Why inhibition of COX-2 is so important in comparison to the first generation drugs, which mainly inhibit COX-1. Please also indicate which serious side effects may COX-2 inhibitors provide. 

We provide more details on parecoxib as follow:

Parecoxib can manage the opioid-induced hyperalgesia [21]. Parecoxib diminishs postsurgical pain and accelerates movement more than controlled analgesia of patient [22]. Parecoxib can multimodal preemptive analgesia in reducing postoperative acute pain in hip and knee replacement patients, and decrease cumulative opioid consumption without increasing the risk of adverse drug events [23]. Parecoxib protects against myocardial ischemia/reperfusion via targeting PKA-CREB signaling pathway [24]. (Please see Introduction section, page 2 and page 3, lines 98-104)

We response the question as follow:

The first generation non-steroid anti-inflammatory drugs (NSAIDs), which mainly non-selective to inhibit both COX-1 and COX-2. Constitutive COX-1 is considered to mediate prostaglandin-dependent gastric protection [18]. COX-1 can assist the secretion of the gastric wall mucosa and prevent gastric acid from erosion the gastric wall mucosa [18]. Therefore, first-generation NSAIDs often have the serious side effect of gastric ulcers [19]. Selective inhibition of COX-2 is so important in comparison to the first generation NSAIDs, because the gastric ulcers rarely occur. The serious side effects provided by COX-2 inhibitors is increased risk of suffering myocardial infarction and death [20]. Rofecoxib was withdrawn from the market for this reason, but the similar COX-2 selective etoricoxib has replenished it in Europe but not in the United States [20]. (Please see page 2, the last paragraph, lines 87-96)

  1. lines 118 - was it trans-resveratrol?

Yes, it is trans-resveratrol. (Please see 2.1. Reagents and Chemicals section, page 4, line 156)

  1. The research is interesting, but the conclusions are to general. The Authors sholud write significant limitations of the study. It is only in vitro study and many various body, diet and drug-dependent factors may have significant influence on such interactions. The Authors should clearly described, that this study has only preliminary significance and much more research (especially in vivo studies) are needed to fully elucidate importance of combining parecoxib with resveratrol. 

Thank you for your suggestion. We add the significant limitation of the study in the conclusions section as follow:

The significant limitations of the study are only in vitro study and many various body, diet and drug-dependent factors may have significant influence on such interactions. This study has only preliminary significance to point that parecoxib and resveratrol combined treatment may offer a hopeful future for colorectal cancer patients and much more research (especially in vivo studies) is needed to fully elucidate importance of combining parecoxib with resveratrol in the future. (Please see Conclusions section, page 15, line 520-526)

Round 2

Reviewer 1 Report

Comments and Suggestions for Authors

The revised manuscript is suitable for publication now.

Comments on the Quality of English Language

n.a.

Author Response

Thank you for your suggestion.

Reviewer 3 Report

Comments and Suggestions for Authors

The Authors have correctly addressed all my questions and the manuscript was significantly improved. There are still some small things, which need to be improved before the manuscript will be processed further:

1. line 50 - if we are discussing natural, dietary sources of resveratrol, we should use the term consumption, because the term "dose" should be used in pharmacological or toxicological issues. So please correct the phrase "beneficial dose" to "beneficial intake"

2. line 55 - "daily uptake" - it sholud be daily intake. Please correct. 

3. lines 57-59 - at the beginning the Authors worte that oral absorption is around 1% and later that "Extensive metabolism in the liver and intestine causes an oral bioavailability considerably lower than 1%"? I see significant discrepancy here. The molecule is first absorbed, distributed and later metabolise in the liver. This part should be re-written, because it sounds a little illogical to me.

4. lines 63-68 - molecules are first absorbed, later distributed throughout the body and finally metabolise in the liver and at the end excreted. Unless, there is a first passing effect, then the molecule before it is distributed is firstly metabolised in the liver. Is there such a mechanism in the case of resveratrol? If so, it should be clearly stated in this part of the manuscript.  

Author Response

Response to Reviewer 3:

We sincerely thank you for your constructive and valuable comments that helped us improve our manuscript. Herein, we provide our point-by-point responses to the comments along with a description of all the changes that have been made and highlighted in green in the revised manuscript.

The Authors have correctly addressed all my questions and the manuscript was significantly improved. There are still some small things, which need to be improved before the manuscript will be processed further:

  1. line 50 - if we are discussing natural, dietary sources of resveratrol, we should use the term consumption, because the term "dose" should be used in pharmacological or toxicological issues. So please correct the phrase "beneficial dose" to "beneficial intake"

Thank you for your suggestion. We have corrected the phrase "beneficial dose" to "beneficial intake". (Please see line 50)

  1. line 55 - "daily uptake" - it sholud be daily intake. Please correct. 

Thank you for your suggestion. We have corrected the "daily uptake" to " daily intake ". (Please see line 55)

  1. lines 57-59 - at the beginning the Authors worte that oral absorption is around 1% and later that "Extensive metabolism in the liver and intestine causes an oral bioavailability considerably lower than 1%"? I see significant discrepancy here. The molecule is first absorbed, distributed and later metabolise in the liver. This part should be re-written, because it sounds a little illogical to me.

Thank you for your suggestion. To avoid illogical, we delete the sentence “The oral absorption of resveratrol in human body is approximately 75% and is believed to take place chiefly through transepithelial diffusion.”. We retain the sentence “Extensive metabolism in the liver and intestine causes an oral bioavailability considerably lower than 1%.”. It is quoted from reference [6]. (Please see lines 56-57)

  1. lines 63-68 - molecules are first absorbed, later distributed throughout the body and finally metabolise in the liver and at the end excreted. Unless, there is a first passing effect, then the molecule before it is distributed is firstly metabolised in the liver. Is there such a mechanism in the case of resveratrol? If so, it should be clearly stated in this part of the manuscript.  

The major sites of resveratrol metabolism include the intestine and liver [6]. There is no first passing effect. The resveratrol is absorbed in the small intestine and sent to the liver for phase II enzyme metabolism and to produce glucuronides and sulfates of resveratrol [6,7]. (Please see lines 64-67)
